# Separation Performance of Membranes Containing Ultrathin Surface Coating of Metal-Polyphenol Network

**DOI:** 10.3390/membranes13050481

**Published:** 2023-04-29

**Authors:** Hluf Hailu Kinfu, Md. Mushfequr Rahman

**Affiliations:** Helmholtz-Zentrum Hereon, Institute of Membrane Research, Max-Planck-Straße 1, 21502 Geesthacht, Germany

**Keywords:** metal–polyphenol network, tannic acid, tannic acid–metal surface coating, TFC membranes, TA–metal self-assembly, coordination

## Abstract

Metal–polyphenol networks (MPNs) are being used as versatile coatings for regulating membrane surface chemistry and for the formation of thin separation layers. The intrinsic nature of plant polyphenols and their coordination with transition metal ions provide a green synthesis procedure of thin films, which enhance membrane hydrophilicity and fouling resistance. MPNs have been used to fabricate tailorable coating layers for high-performance membranes desirable for a wide range of applications. Here, we present the recent progress of the use of MPNs in membrane materials and processes with a special focus on the important roles of tannic acid–metal ion (TA-M^n+^) coordination for thin film formation. This review introduces the most recent advances in the fabrication techniques and the application areas of TA-M^n+^ containing membranes. In addition, this paper outlines the latest research progress of the TA–metal ion containing membranes and summarizes the role of MPNs in membrane performance. The impact of fabrication parameters, as well as the stability of the synthesized films, is discussed. Finally, the remaining challenges that the field still faces and potential future opportunities are illustrated.

## 1. Introduction

Thin-film composite (TFC) membranes, with less than a 50 nm selective layer [1], are mostly fabricated over a porous support via an interfacial polymerization (IP) reaction at the interface between two immiscible phases. TFC membranes have been extensively utilized in diverse separation processes owing to their high solute rejection and a wide pH tolerance [1,2]. Despite the maturity of the state-of-the-art IP process, the addressing of several practical and fundamental issues is still required.

The membrane manufacturing process involves a large amount of toxic organic solvents posing a series of health and environmental risks [3]. In fact, the production of polymeric membranes generates 50 billion liters of wastewater that contains harmful solvents every year [4]. Furthermore, after fabrication, the release of residual solvents during the membrane lifecycle remains an issue [5]. Therefore, it is critical to stimulate the search for alternative ecofriendly methods. Addressing the sustainability concerns related to membrane fabrication and materials is a developing field of study [1]. Xie et al. [6] proposed five strategies to improve the sustainability of the membrane manufacturing process: polymers from renewable sources, greener solvents, recycling and treating wastewater generated during fabrication, reducing fabrication steps, and dissolving casting solution at room temperature to save energy.

In a quest for sustainable thin film membrane fabrication, coating approaches using aqueous solutions have been practiced for preparing non-polyamide thin films. They are simple, cost-effective, and ecofriendly processes through which one can easily control film thickness and structure. However, these procedures suffer from low permselectivity principally because of the increased film thickness compared to PA-TFC membranes [7].

Recently, the emergence of polyohenol-based, such as polydopamine and tannic acid, TFC membranes have shown promising advancements in sustainable membrane synthesis procedures [8,9,10]. The green synthesis method and anti-fouling property of the membranes have attracted considerable attention. Tannic acid is a highly water-soluble [1] natural plant polyphenol found in plants such as oak, tea, and fruits. One of the important features of tannic acid is its ability to adhere robustly onto various substrates. These include hydrophilic and hydrophobic, organic and inorganic, particle or surface substrates [11]. Adhesion occurs through different or a combination of many interactions such as hydrophobic interaction, hydrogen bond, electrostatic interactions, and coordination [12,13]. Metal chelation is a salient feature of many polyphenols. Polyphenols act as organic polydentate ligands, whilst transition metal ions play a role as crosslinkers. The coordination between tannic acid and transition metal ions, e.g., ferric ion, results in robust conformal films as reported in a recent study [14]. Natural polyphenols coordinate with metal ions to form MPN complexes through coordination-led self-assembly. Tannic acid, a typical natural polyphenol, has the ability to form metal–ligand complexes by chelating many transition metal ions such as Fe^3+^, Cr^3+^, Co^2+^, Zn^2+^, Zr^4+^, Mn^2+^, Cu^2+^, etc. [14,15,16,17]. Iron is the most studied metal for self-assembly with TA. At optimum conditions, a coordination reaction between catechol or galloyl groups and iron ions forms a stable octahedral complex [14]. However, this coordination is extremely pH dependent as the number of organic ligands attached to the metal ion decrease down the pH range. Three complex formation states are mainly recognized, as shown in Figure 1. However, it is worth noting that the ligand-to-metal ratio, the final concentration, and the mechanism of film formation influence the coordination complex state [18,19].

TA–metal ion complexes have diverse applications in metal adsorption, water and wastewater treatment, drug delivery, nano-encapsulation, and many more [20,21]. A newly introduced application area for the polyphenol coordination with metals is in the topics of membrane technology. Tannic acid–metal ion (TA-M^n+^)-based MPN networks are used both for coating of membrane surfaces [11,22,23] and as a selective layer over a support [7,14,24]. For selective layer synthesis, the foundation lies on polyphenol’s ability to complexate with metal ions though coordination reaction or, particularly, supramolecular self-assembly. This self-assembly results in a thin film of a few nanometers that can be used for the selective separation of components. Tannic-acid-containing membranes have high water permeability because of the superhydrophilic surfaces as a result of its abundant hydroxyl groups enveloping the polyphenol alongside sufficient rejection of many salts and organic components [7,11,25]. This allows synthesis of films meeting the demands breaking the ubiquitous permeability–selectivity tradeoff. These membranes have been utilized in removal of trace organic contaminants [26], rejection of heavy metals and pharmaceuticals [27,28], nanofiltration [7,8], preparation of antifouling and antibacterial membranes [27], forward osmosis [29], and dye separation [25]. They are also shown to be efficient in oil/water separation with their hydrophilic, superoleophobic, and antifouling properties [30]. The nanofilm can be formed over various substrates in a simple, green, and fast facile technique. Nevertheless, this TA−M^n+^coordination-based membrane synthesis technique is not yet a decade in age.

Here, we present the recent progress in TA−M^n+^−based membrane development. This work discusses the different practices coordination chemistry has been used for membrane film synthesis, methods of TA-M^n+^ membrane fabrication, factors affecting the thin film formation, and the latest application areas of MPN-based membranes investigated in several studies. Furthermore, we highlight the major challenges for real-life application of these thin films. Emphasizing the progress made, together with the prospects put in this paper, we aim for making significant improvements in the membrane fabrication and separation processes through future studies.

## 2. Polyphenol–Metal-Assembly-Based Membranes

The use of TA for the fabrication of nanofiltration (NF) membranes is an emerging trend. On one hand, tannic acid is used to find an alternative selective layer; on the other hand, it is also used to prepare an intermediate layer (often termed as a gutter layer or interlayer) between the porous support and the polyamide layer of NF membranes. The presence of catechol and gallol groups in TA provide the utilization of both non-covalent and covalently bonded layers on top of the porous substrates [24]. While TA and trimesoyl chloride (TMC) have been used as monomers for interfacial polymerization monomers to prepare TFC NF membranes [31], environment-friendly casting of TA–M^n+^-coordinated layers are the most popular choice. The use of an organic solvent, which is immiscible with water, is mandatory for interfacial polymerization. *n*-Hexane, a popular choice, has several concerns over health and ecological hazards. The benefit of metal–polyphenol-based membranes is that both tannic acid and metal constituents are dissolved in aqueous solution. Furthermore, the fast, one-step assembly using the natural low-cost polyphenol simplifies small, as well as large-scale, membrane fabrication processes. Assembled selective layers significantly improve membrane characteristics. It is reported that TA-Fe^3+^ complex thin film membranes can go up to a small average pore size of 0.7 nm with a narrow pore size distribution, which helps to realize high-precision separation [28]. Film thickness varies from a few nanometers to 30 nm. This layer is synthesized through different coordination methods via the self-assembly approach. Supramolecular self-assembly between metal ions and the organic ligand allows the fabrication of NF membranes without the use of reprotoxic organic solvents [7,8,28].

To categorically assess MPN-based membranes, here it is reviewed that tannic acid can be utilized for membrane film synthesis in three main ways depending on the chemistry and purpose of modification: surface modification of membranes, interlayer construction and selective layer synthesis.

### 2.1. Membrane Surface Modification

Tannic acid is used for surface modification of NF, ultrafiltration (UF), and microfiltration (MF) membranes through sequential deposition or co-deposition with a metal ion. The self-assembly between TA and Fe^3+^ is especially studied for coating and surface modification of various membranes [8,23,26,27,32]. The main advantages of this surface modification coating material can be summarized as bringing improved surface hydrophilicity leading to decreased surface transport resistance and, thereby, higher water permeance [33,34], improved anti-fouling [27,35], and a simple, easy, fast surface modification procedure [27]. Polyamide membranes are prone to permeance loss owing to degradation from exposure to chlorine in water [36]. TA-M^n+^ networks enhance chlorine resistance by reducing the chlorine-sensitive sites. Moreover, the strong free-radical scavenging effect of TA helps to increase membrane chlorine resistance by lowering chlorine radical levels [35]. The coating of surfaces to enhance permeability is associated with the profusion of hydroxyl groups in the polyphenol.

Modification of membrane surface properties with TA-M^n+^ tackles the main challenges of membrane filtration: organic-, oil-, and bio-fouling and membrane contamination by microorganisms [27]. Furthermore, surface modification of the membrane active layer with MPN films enhance membrane performance. It was reported that TA-Fe^3+^ regenerable coatings on a polyamide layer increases salt rejection, as well as divalent to monovalent ion selectivity [23]. Guo et al. [26] showed that this facile and fast coating procedure can significantly improve the rejection performance of organic contaminants that cannot be effectively retained by commercial NF membranes. Moreover, TA–metal complex layer coatings can enhance the selectivity of MXene membranes [37]. Surface coating of the 2D MXene film using tannic acid self-assembly with iron, copper, or zinc imparts enhanced contaminant recovery, such as dye molecules and hydrophobic micro-pollutants. This opens a new simplistic strategy for adjusting the selectivity and efficiency of numerous types of membranes.

### 2.2. Interlayer Construction

The use of polyphenols such as polydopamine and TA as a bridging agent (i.e., gutter layer) between porous supports and a selective layer from interfacial polymerization has emerged in the last decade. The primary goal of phenolic interlayers is in increasing the permselectivity. Polyphenol-containing interlayers facilitate the wetting of the substrates by the aqueous solution and control the diffusion of the component in the aqueous solution, such as piperazidine (PIP), during the IP process [38]. Customary TA−containing interlayers involve co-deposition of TA with a crosslinker or reactant such as diethylenetriamine [38], polyethyleneimine (PEI) [39], dual diazonium salt (DDS) [40], or transition ion metals. TA−Fe^3+^intermediate layer, for instance, brings high surface hydrophilicity and controls adsorption/diffusion of the amine monomer during IP [41]. Furthermore, it allows the construction of a smooth, ultrathin, dense PA surface structure. The TA-Fe^3+^ interlayer containing membranes exhibited high separation performance and solvent permeability, presenting a potential for organic solvent nanofiltration application [41].

TA–metal ion complex interlayers allow the preparation of defect-free thinner active-layered polyamide TFC membranes [38,42]. A triple-layered TFC membrane was fabricated through IP of the reaction between m-phenylenediamine (MPD) and trimesoyl chloride (TMC) over TA-Fe^3+^-coated PVDF microfiltration support [29]. The TA-Fe^3+^ interlayer not only prompted a thinner polyamide layer but also had an impact on membrane hydrophilicity, pore structure, and surface roughness. Water flux was tripled compared to the TFC membrane without the TA-Fe^3+^ interlayer. The proposed synthesis technique ensures high controllability of chemical and physical structures and scale-up prospects for industrial applications. In a similar investigation, Yang et al. [42] constructed a TA-Fe^3+^ nanoscaffold layer over a polysulfone substrate prior to interfacial polymerization of TMC and piperazine (PIP), as shown in Figure 2a. Due to its smaller pore size, the MPN-based interlayer prevents intrusion of polyamide into the pores of the support substrate. Moreover, the TA-Fe^3+^ interlayer enhances uptake of amine monomers in addition to regulating their controlled release. The MPNs’ interlayer containing TFC membrane showed a water permeance of 19.6 L∙m^−2^∙h^−1^∙bar^−1^, while that of the control TFC membrane was 2.2 L∙m^−2^∙h^−1^∙bar^−1^. Rejection of salts was also improved, and the membrane performance surpassed the permeance versus selectivity upper bound.

Metal–polyphenol network interlayers can also act as metal precursors for metal–organic framework (MOF) membrane synthesis. The ultrathin MPN top layer of TA-Zn^2+^ deposited over a polyethersulfone (PES) substrate surface in a layer-by-layer fashion demonstrated itself to be a promising metal precursor in the fabrication process of ZIF-8 membrane [43]. Zn^2+^ is the main metal precursor in ZIF-8 membrane synthesis. The MPN-coated PES substrate was immersed into a 2-methylimidazole aqueous solution. Then, an ultrathin MOF top layer is constructed because of the partial self-conversion of the TA-Zn^2+^ network to ZIF-8. The schematic representation of ZIF-8/(TA-Zn^2+^)n/PES membrane synthesis is shown in Figure 2b,c. This process provides sufficient controllability of the MOF membrane synthesis. The synthesized nanofiltration membranes showed excellent inorganic salt rejection, which can be used for desalination purposes.

### 2.3. Tannic Acid-Metal Ion Based Selective Layer

A thin layer of TA–metal complex deposited over a porous support substrate can act as a separating layer in many membrane technology processes. The coordination of TA with metal ion for active layer synthesis is a relatively new research topic and much of the membrane research conducted on metal–polyphenol-complex-based films are on TA-Fe^3+^. However, significant studies have been undertaken utilizing other transitional metal ions. You et al. [44] examined the membrane performance behavior of NF membranes synthesized using various crosslinkers. It was revealed that characteristic membrane properties could be regulated through a vast choice of metal ions. TA-Ag^+^, TA-Cu^2+^, and TA-Fe^3+^ NF films exhibited a relatively low water permeance compared to TA-Co^2+^- and TA-Ni^2+^-assembled films. At a 1:1 ratio of tannic acid to metal ion, the membranes displayed a pure water flux in the range of 8–47 L∙m^−2^∙h^−1^∙bar^−1^ while maintaining a more than 94% rejection of methylene blue (MB), as shown in Figure 3c. Interestingly, the variation in molar ratio between tannic acid and Ni^2+^ was used to fine tune the membrane performance. The hydrophilicity of the TA–metal membranes can also be regulated by the variation of the TA-to-metal ratio, as shown in Figure 3d. The increase in Ni^2+^ content raises the membrane surface water contact angle. This is due to the consideration that more phenolic hydroxyl groups in TA interact with Ni ions [44]. As a result, the number of hydrophilic groups are reduced, decreasing the hydrophilicity of the membrane surface. Guo and co-workers [9] synthesized a TA-Fe^3+^-selective layer with high rejection of hydrophobic organic contaminants outperforming commercially available NF membranes. Moreover, the prepared non-polyamide selective layer exhibited a high water/organic molecule selectivity. This provides an insight into the fabrication of active thin films using the coordination chemistry as an alternative to the prevalent organic-solvent-consuming TFC membrane synthesis strategies.

Wang et al. [30] reported an in situ technique of preparing superwetting membranes for oily wastewater treatment though TA–metal assembly. They have proven that those metal ions with a high ionization potential, according to the soft–hard acid-base theory (HSAB), has a higher coordinating affinity with tannic acid to form stable complexes. Concerning the morphological characteristics, Cu^2+^, Zr^4+^, and Fe^3+^ based membranes showed rough surfaces with larger pore sizes compared to membranes containing Ni^2+^, Co^2+^, and Fe^2+^. This provides a broad choice of coordination chemistry for synthesizing MPN-based membranes. For example, TA-Ni^2+^ and TA-Cu^2+^-based membranes show a higher complex stability [30]. In addition to surface morphology, membrane characteristics such as surface wettability, flux, and rejection of TA–metal ion thin films are affected by the choice of the complexing metal ion centers. In another study, Chakrabarty et al. [45] synthesized a thin skin layer of coating from TA and Cu^2+^ ions’ coordination reaction over PAN membrane support to prepare a high-flux NF membrane. Deposition of ethanolic solutions of Cu(CH_3_COO)_2_ and TA to the surface of the PAN support resulted in the formation of a stable thin layer. The membrane maintained a rather constant flux for several days, showing an excellent adhesion behavior between the selective complex thin layer and the PAN support, implying good mechanical stability.

In the coordination chemistry, it is essential to utilize the adhesion, adsorption, and electrostatic interactions between the support, the polyphenol, and metal ions. Consideration of molecular interactions allow the regulation and formation of defect-free thin film. Liu et al. [7] used hydrolyzed PAN porous support to prepare TFC membranes containing a TA-Fe^3+^-selective layer, as shown in Figure 3a. The electrostatic and coordinative affinity of Fe^3+^ with the –COOH-rich support layer facilitated the formation of a homogeneous TA-Fe^3+^-selective layer and further improved the structural stability of the membrane, as shown in Figure 3b. A comprehensive investigation on the as-synthesized membrane morphology and property revealed that the structure and performance of the selective layer can be controlled by the fabrication conditions. Short coordination time of a few minutes was adequate for the formation of the self-assembled TA-Fe^3+^ layer. Further increase in assembly duration brought no significant change. This rapid fabrication process makes it an ideal strategy for large-scale membrane production process. Moreover, a variation in pH of monomer solutions showed that a slightly alkaline condition is ideal for the formation of a dense TA-Fe^3+^-selective layer because the tris-complex of tannic acid and iron ion is established under an alkaline condition [7].

A much-detailed study on the effect of fabrication conditions on the synthesis of superhydrophilic membrane was performed by Xiao et al. [25]. The proposed method of layer-by-layer assembly of TA and Fe^3+^ realized a facile process of fabricating multilayered crosslinked metal-polyphenol networks. To prevent the precipitation of metal ion, TA molecules, or TA-M^n+^ complexes as aggregates in water before depositing over a support surface, Xiao et al. [25] used a layer-by-layer assembly for preparing a defect-free TA-Fe^3+^-selective layer. An MPN TFC prepared at a pH of 8 of precursor solutions resulted in the most compact selective layer with the highest retention. Both bilayer number and assembly time increase, resulting in a decrease in water flux. Similar thin selective layers for NF application were also synthesized via LBL technique in another study [28]. Alternate deposition of low concentrations of TA and Fe(NO_3_)_3_·9H_2_O solutions produced ultrathin and defect-free multilayered nanofilms. The tailored pore size of less than 1 nm confirmed that the films can be used for any NF purpose.

Another novel work on the TA–iron network for osmotically driven process of forward osmosis (FO) was reported recently [46]. In this study, a separating top layer of 20–30 nm was synthesized over a UF support and displayed a water flux as high as 14.2 L∙m^−2^∙h^−1^∙bar^−1^. Moreover, the FO film ensured more than three orders of magnitude selectivity to sunset yellow dye than for NaCl, outpacing a commercial TFC FO membrane from HTI in both rejection and permeability. These polyphenol-network-based membranes are applicable in resource recovery processes where salt rejection is not a primary target. These include pretreatment of seawater to minimize biofouling and scaling risks for RO process, urine treatment for retention of nitrogen and phosphorous, removal of micro-pollutants from surface water, resource recovery in wastewater, and dye retention. FO-based separation using metal–polyphenol network membranes are especially applicable for non-desalination purposes where polyamide membranes with high salt retention and accumulation are not suitable. MPN-based active layers also prevent the decline of water flux caused by concentration polarization during operation. Table 1 lists selective layers used for membrane application based on the self-assembly between transition metal ions and tannic acid.

## 3. Synthesis Strategies of Metal–Polyphenol Membranes

Metal–organic ligand networks are spontaneously assembled from mixtures of tannic acid and metal ion solutions. Metal-coordinated tannic acid networks grow in all orientations. However, a well-structured, highly permeable, and selective thin film could be synthesized through controlled and systematic assembly. This requires a proper synthesis procedure. So far, the two widely used techniques for tannic acid–metal thin film membrane synthesis are co-deposition and layer-by-layer self-assembly. Nevertheless, interfacial polymerization (IP) has also been applied [24]. TA-Fe^3+^-coordinated network fabrication was attempted through IP using an aqueous solution of TA dissolved in water and the organic phase of tris(acetylacetonat)iron(III), Fe(acac)_3_ dissolved in n-hexane. However, synthesis of metal–polyphenol networks through IP is uncommon. More importantly, IP requires dissolution of one of the reagents in toxic organic solvent.

One of the facile and fast synthesis techniques of metal–polyphenol networks is co-deposition. Co-deposition is often referred to as one-step self-assembly of metal and polyphenol. The top layer of a microfiltration or ultrafiltration membrane supports are exposed to the aqueous reagent solutions containing the polyphenol and metal ion, respectively. In this technique, one of the reagent solutions is poured over the membrane support shortly after the other [7,9,45], or the support is immersed in a fresh mixture of the reagents [44]. A schematic representation of co-deposition is shown in Figure 4a. A one-step process of a thin skin layer formation through coordination reaction was reported using ethanolic solutions of TA and copper(II) ions [45]. Reagent solutions are poured into a PAN support mounted between a Teflon plate and frame in 10 s difference and allowed to react for 10 min. Through the coordination process between Cu^2+^ and phenolic groups in TA, a remarkably stable film with around 600 Da molecular weight cut-off (MWCO) is formed. Gou et al.’s [9] research is one of the pioneering works on metal–polyphenol-based non-polyamide NF membranes. MPN-based film was synthesized via rapid assembly over a porous poly(ether sulfone) (PES) support in a short reaction time (less than 2 min). The obtained TA-Fe^3+^-based membrane not only improved the rejection of trace organic contaminants but also showed a potential breakage of the permeability–selectivity trade-off. Although they exhibit a high water-permeation property, metal–polyphenol-based membranes synthesized via co-deposition show low salt rejection, which is not sufficient enough for high-performance nanofiltration. Recent research optimized the rapid one-step self-assembly of TA and Fe^3+^ and obtained a TFC membrane with MWCO of ~390 Da. The TFC membrane showed high rejections for salts in a sequence of Na_2_SO_4_ (90.2%) > MgSO_4_ (83.4%) > NaCl (50.0%) > MgCl_2_ (35.2%) [7]. Moreover, it exhibited high rejections for organic pollutants (e.g., >99.0% dyes, 92.2% streptomycin, and 81.8% chloramphenicol), while maintaining a water permeance as high as 13.6 L∙m^−2^∙h^−1^∙bar^−1^. The one-step assembly of polyphenol–metal-coordinated networks is not only used to prepare the selective layers of TFC membranes, it has also been widely used for surface modification of the membranes [8,26,27,33]. The assembly time specified for the polyphenol and metal ion mixture plays a significant role in determining the morphology of the MPN layer coated by co-deposition. Additionally, membrane structure and separation performance of the assembled membrane can be regulated and enhanced through optimization of other factors, which are discussed in Section 4.

A polyphenol–metal layer can also be coated by sequential deposition of tannic acid and metal ion solutions over a substrate, layer-by-layer (LBL) self-assembly. LBL is ideal for the preparation of ultrathin selective layers. Technically, a porous membrane support is alternately immersed into pre-prepared solutions of polyphenol and metal ion, as shown schematically in Figure 4b. This process is analogous to the synthesis of polyelectrolyte multilayer films. LBL allows better control over the surface porosity and pore size of the membranes compared to co-deposition [28]. The thickness of the metal–polyphenol network increases with the number of deposited layers. LBL mitigates the challenge of loose network formation. In co-depostion, a loose polyphenol–metal network can be formed because of the formation of Fe^3+^ aggregates, which cannot penetrate the growing TA-Fe^3+^ layer to crosslink TA molecules. This results in a highly porous, in some case, defected metal–tannic acid layer. Lin et al. [28] prepared defect-free NF membranes containing a metal–tannic acid-coordinated selective layer having an average pore size of 0.7 nm through the LBL technique. A coordinated bilayer, two layers of TA, and two layers of ferric ion deposited alternately had a narrow pore size distribution and a thickness of only 5 nm. A water permeance of 12.4 L∙m^−2^∙h^−1^∙bar^−1^ was reported and exceeded the upper bound in the permeability–selectivity plot compared to other nanofilms of various materials, as shown in Figure 5. Another series of multilayered thin films for the NF process were synthesized facilely through an LBL self-assembly strategy [25]. PAN UF membrane was immersed in a certain amount of TA solution for certain time and rinsed with DI water to remove any unstable unattached TA molecules. Then, the TA-coated support was subjected to a definite concentration of Fe^3+^ solution, and the process was repeated. The resultant membrane was endowed with high flux and good selectivity toward several dyes, displaying a significant promise in the field of molecular separation. The LBL technique enhances surface hydrophilicity by increasing the bilayer number and subsequent introduction of abundant phenolic groups [25]. Moreover, MPN membranes synthesized via LBL show superior rejection compared to co-deposited film [25].

An uncommon method of MPN membrane synthesis is contra-diffusion. Contra-diffusion allows the construction of thin layers through coordination restricted within a confined space. Unlike in the co-deposition and LBL techniques, this method prevents undesired thickening of the active layer in the TA-M^n+^ TFC membrane by regulating the competition between the diffusion of precursors [48]. Fe^3+^ diffuses faster than TA because of their size differences. Thus, Fe^3+^ ions diffuse through the porous structure of the membrane and coordinate with TA at the membrane active-layer side. As a result, a defect-free selective layer is generated. A schematic representation of the aqueous contra-diffusion method is shown in Figure 4c.

Additionally, MPN films can be deposited over porous support substrates through solvent/nonsolvent exchange during a phase inversion process for membrane fabrication [30]. The membrane is fabricated via the nonsolvent vapor-induced and liquid-induced phase separation. A casting solution containing TA is cast and exposed to humid water vapor. Subsequently, the obtained film is immersed in an aqueous metal ion solution to induce precipitation, as well as coordination. This technique prevents surface pore blockage from the rapid complexation of TA and metal ions, unlike in co-deposition or LBL. Moreover, it allows surface modification together with control of pore structures and membrane morphologies, which is challenging when using an already prepared commercial support. MPN-incorporated PES membrane synthesis through NIPS was also reported [32]. TA-Fe^3+^ complexes were employed as additives to regulate membrane surface properties. However, the dispersion of TA-M^n+^ networks in the integral membrane structure signifies an increase in the selective layer thickness. Besides, the phase inversion mechanism requires the use of organic solvents.

## 4. Factors Affecting Metal–Polyphenol Complex Formation, Membrane Selective Layer Characteristics, and Performance

### 4.1. Effect of Concentration and TA to Metal Ion Ratio

Monomer concentration has a significant effect on the growth of thin films generated by self-assembly or coordination reaction [27]. Thicker and more compact coatings are generated with a higher monomer concentration, which increases the degree of coordination reaction. Moreover, an increase in monomer concentrations results in a water contact angle decline and, thereof, increases the hydrophilicity of the assembled films [7]. The hydrophilicity enhancement is mainly attributed to TA hydroxyl group abundance. Membranes containing the TA-Fe^3+^-selective layer from the same concentration of both TA and ferric ion (0.1–0.4 wt%) showed this distinctive feature [7] and were able to remove some portions of inorganic salts from the solution. As in negatively charged membranes [49], the order of salt rejection ratios followed Na_2_SO_4_ > MgSO_4_ > NaCl > MgCl_2_ [7]. The most important finding in this study is that with the increase in TA and Fe^3+^ concentrations in the coating solutions, salt rejection ratios of the assembled membranes are increased initially and decreased afterward. The pure water flux values, as anticipated, change in the opposite direction. A membrane with 0.3 wt% showed the highest salt rejections (i.e., 90.2% Na_2_SO_4_, 80.2% MgSO_4_, 50.0% NaCl, and 35.2% MgCl_2_), whilst simultaneously maintaining a pure water permeance as high as 67.8 L∙m^−2^∙h^−1^ at 0.5 MPa. A proper explanation for this phenomenon is that the selective layer is not dense enough at low concentrations of monomer solutions. Membrane porosity decreases as the monomer concentrations increase. Conversely, after a certain point, excess reagents provoke rapid aggregation of TA-Fe^3+^ complexes at the membrane surface, leading to the development of particles, wrinkles, and defects in the constructed thin film [7]. In TA–copper ion self-assembled membranes, the increase in metal concentration results in a continuous decrease in water permeability [45]. A more concentrated metal ion solution can bind more tannic acid monomers on the membrane support surface, leading to thicker coatings.

Maintaining a constant concentration of TA and varying ferric ion concentration leads to a decrease in water permeance with the increase in Fe^3+^ concentration [26]. Dense and highly crosslinked thin layers with a low water permeance and high rejection toward salts and organic molecules are synthesized at a higher Fe^3+^ concentration. However, a further increase in ferric ion results in a decrease in salt rejection and increase in pure water flux. Another study by Fan et al. [8] involved the variation of TA concentration while maintaining Fe^3+^ concentration constant. SEM images revealed that membrane surfaces became smoother and compact with an increase in TA concentration. Furthermore, at a Fe^3+^ concentration of 1 g/L, an increase in TA concentration from 0.5 to 1.5 g/L brought dramatic changes in membrane performance. The water flux evidently declined from 161.4 to 45.6 L∙m^−2^∙h^−1^, while orange II rejection increased from 71.5% to 94.8%, and the Na_2_SO_4_ rejection ratio increased from 20.6% to 62.1%, respectively. A further increase in TA concentration up to 3 g/L did not lead a substantial variation of flux and rejection values.

### 4.2. Effect of Solution pH

Acidity or alkalinity of the precursor solutions govern the complex formation states of metal–polyphenol self-assembly. Membranes synthesized at various pH values exhibit diverse morphology characteristics. With an increase in the pH value of monomer solutions, an increase in the protrusions size on membrane surfaces can be observed [25]. Oxidation of catechol groups into reactive quinones is accelerated at high pH. Quinones undergo a self-crosslinking reaction and contributes to the TA aggregate formation (Figure 6i). TA aggregates result in an increase in protrusion size on the surfaces of membranes. In contrast, at low pH values, protonation of hydroxyl groups is increased. This results in the formation of a TA–metal ion mono-complex and thinner selective layers. A rise in pH induces the ionization of the tannic acid functional groups (Figure 6i). This boosts the coordination self-assembly and leads to a dense top layer. In general, an increase in pH changes the complex formation from the mono- to bis- to tris-complex state (Figure 1). Additionally, an increase in solution pH also results in an increase in the thickness of the synthesized film, as shown in Figure 6a–h. Nonetheless, in the case of ferric salt solutions, at a pH of 9 or more, generation of ferric hydroxide is facilitated [42]. This phenomenon interferes with the assembly of TA-Fe^3+^ and leads to a reduction in both film thickness and rejection [8,25]. A significant decrease in the water contact angle at the surface of the membrane is reported when the pH of the casting solutions is increased [25]. This exhibited a superhydrophilic property, increasing the interaction between membrane pore walls and water molecules and enhancing water permeability by increasing the infiltration capillary force [50].

### 4.3. Assembly Time

The influence of assembly time on MPN film formation is inevitable. However, assembly time has a small influence on membrane performance, indicating that the coordination is instantaneous and could not be enhanced significantly with extended coordination time [8]. It was noticed that for tannic acid and Cu(II)-complex-based film synthesis, an increase in coating time showed a water permeability decrease until a steady state is reached and no further flux decline occurred [45]. As investigated by several studies, TA-metal ion complex formation is a fast coordination reaction. One minute was enough to fabricate a dense selective layer of NF membrane from TA and ferric ion complexion [25]. Further prolonging of assembly time only slightly enhanced dye rejection whilst significantly decreasing water permeance. Liu and colleagues [7] also confirmed that less than two minutes was adequate for preparation of an assembled selective layer of the TFC membranes for the removal of salts.

In addition to coordination time, storage time of the precursor solutions influences final membrane structure and performance. Iron ion aggregation not only alters the chemical composition (TA-to-metal-ion ratio) of the synthesized films but also the physical structure of the TA-Fe^3+^ complex [28]. Prolonged storage of iron solution results in aggregation, and hence, defects and large pores are developed in assembled film, as shown in Figure 7c. With an increase in solution storage time, the ratio of Fe^3+^ to TA in the complex increases. Fe^3+^ aggregates are too large to penetrate into the growing TA films. This results in a much-loose film and large gaps (defects) between uncoordinated TA molecules and the TA-Fe^3+^ complex containing aggregates. Therefore, usage of fresh metal ion solution is important. These non-aggregated ions allow the Fe^3+^ to be small enough (mono) that it could penetrate into the already deposited TA films and form a crosslinked nanofilm by coordinating with TA molecules. Strict control of aggregation boosts the preparation of defect-free membranes of narrow pore size distribution with enhanced precision of separation.

### 4.4. Number of Deposited Layers

The thickness of assembled nanofilm increases linearly with the number of bilayers deposited specifically in LBL construction. Spectroscopic ellipsometry shows that each TA-Fe^3+^ bilayer is approximately 2.5 nm [28]. A bilayer consists of one TA–metal ion layer. Lin et al. [28] reported one of the thinnest of membranes used for liquid separation using a TA-Fe^3+^ film of two bilayers constructed over PES support. The increase in the number of deposited layers results in an increase in the thickness of the assembled selective layer. As a result, flux decline is observed. Research performed on self-assembly through the layer-by-layer technique involves ferric ion, and the effects associated with multilayers involving other metals are as yet unknown. Xiao et al. [25] prepared TA-Fe^3+^-based NF membranes via a layer-by-layer (LBL) self-assembly technique. This method was proposed to eliminate the problems of low atom economy in membranes synthesized via co-deposition. The first layer of TA deposits over the PAN support because of the substrate-independent adhesive property of TA. Then, ferric ions coordinate with -OH groups of the already attached TA on the surface. For the second layer of TA-Fe^3+^ complex film, incoming TA molecules self-assemble with the already deposited Fe^3+^ ions. The process continues until a membrane film of desired characteristics is acquired. They reported a high water permeability of 40.9 L∙m^−2^∙h^−1^∙bar^−1^ whilst maintaining a rejection of more than 93% toward dyes of 320 Da or more molecular weight. With an increase in the number of layers, a decrease in water contact angle to as low as 28° was achieved. It was found that only one bilayer membrane displayed low dye rejections and high water permeance. However, with an increase in the number of bilayers, the dye rejections increase while water permeance declines. At a bilayer number of 5, the resultant membrane exhibited static blue carmine red dye (IR) rejection of 85.9%, a rose red sodium salt dye (RB) rejection of 98.2%, and water permeance of 25.9 L∙m^−2^∙h^−1^∙bar^−1^ relative to 40%, 90%, and 76 L∙m^−2^∙h^−1^∙bar^−1^ at 1 bilayer number, respectively.

### 4.5. The Effect of Ionic Strength

The influence of ionic strength on metal–phenolic networks film deposition has been explored before [51]. An increase in salt concentration leads to the formation of rough and thicker films. High ionic strength results in galloyl groups extending out from the Fe^3+^ center and interact with other TA–metal ions complexes [51]. A further increase in salt concentration results in an increase in surface roughness. At low salt concentration, the TA solution is homogenous. However, when ionic strength is increased, TA particle size extends and the solution becomes turbid [24]. Electrostatic repulsion forces TA molecules to disperse well in aqueous solution. However, with severe increase in salt concentration, repulsion is screened and results are in aggregation of TA molecules and TA-M^n+^ complexes. The effect of ionic strength on TA-Fe^3+^ complex formation in solution is shown in Figure 7a. Individual TA-Fe^3+^ complexes agglomerate and form clusters when the salt concentration is increased. In addition, ligand-to-metal charge transfer (LMCT) depicting the formation of tris-complex of TA-Fe^3+^ increases. In this phenomenon, screening of charges plays an important role in the formation and growth of the film. MPN complexes undergo conformational changes depending on the ionic strength of the solutions [51].

Generally, change in ionic strength affects surface inhomogeneity and thickness because of the screening of charges in solution [51]. This implies deposition of more open and defected films. The effect of ionic strength on metal–organic composite membrane prepared from TA and Fe^3+^ through interfacial polymerization was explored by Shen et al. [24], and it was revealed to have affected the aggregation of TA-Fe^3+^ complexes. By reacting the aqueous solution of TA and iron acetate dissolved in h-hexane, they reported an increase in rejection of VB12 and a decline in permeability with an increase in ionic strength of the TA solution (Figure 7b). Due to the steric effect in the compact complexes at low ionic strength, the phenolic groups in a compact TA-Fe^3+^ complex can only form mono- and bis-complexes. With an increase in salt concentration, the phenolic groups prefer tri-complex formation and result in dense membranes of low porosity [24]. Nevertheless, with a further increase in salt concentration above the threshold level, aggregation dominates and results in large gaps between consecutive TA-Fe^3+^ complex aggregates, and causes the formation of defected selective layers. However, the effect of the ionic strength of monomer solutions on structure, as well as the performance of membranes prepared directly via polyphenol and metal ion self-assembly in an aqueous state, is yet to be explored.

## 5. Membrane Stability and Durability

One factor that could deteriorate membrane performance is fouling. Although many modifications have been made to mitigate fouling, it is still a great hindrance for NF membranes and limits membrane operations competitiveness and cost effectiveness [52]. Polyphenol–metal-complex-based membranes have the potential to safeguard the surface selective layer and control fouling during operations. TA-M^n+^ layers improve the stability and anti-fouling property of membranes through several mechanisms. A hydration layer is formed at the membrane/water interface because of the hydrophilic nature of the surface. This hydration layer repels hydrophobic foulants, such as oils and proteins, from the surface. The abundant hydroxyl groups in the TA molecular structure preferentially absorb H_2_O molecules to form a protective thin water layer on the exposed surface. Anti-fouling performance study of polyphenol–metal-assembled membranes have shown promising resistance to fouling [7,9,27,32]. A continuous filtration experiment with a representative foulant of bovine serum albumin (BSA) to test the anti-fouling performance of TA-Fe^3+^ membrane was performed by Liu and his colleagues [7]. More significant flux recovery enhancement was observed in the self-assembled membranes than in the control polyamide membrane, assuring that flux can be well-restored after hydraulic washing. In addition to the formation of a hydration layer, surface repulsive interaction enhances fouling resistance by TA-M^n+^ selective layers. The negatively charged functional groups of the membrane surface fouls less by foulants exhibiting low isoelectric point [7]. A strong anti-fouling property of these membranes against microorganisms was also observed [27]. These features provide an insight into polyphenol-based metal crosslinked membranes for a wide range of filtration applications.

A strong adhesion confines the assembled metal–polyphenol film on membrane support substrates. The robust affinity of polyphenol–metal-assembled membranes originates from the galloyl and catechol groups [14]. To verify the strength of adhesion between the self-assembled thin film and PES support, an inverse operation performed on several membranes revealed that Na_2_SO_4_ rejection remains unchanged in TA-Fe^3+^ membranes, while it declines from 95.2% to 88.4% for commercial control PA membrane [7]. Figure 8a shows a typical example of a long-period filtration test for stability of the TA-Cu^2+^ membrane. Polyphenol–metal assembly-based membranes and their performance under long filtration operation are summarized in Table 2. It presents MPN-based selective layers displaying stable filtration performance and possessing decent durability. Metals of high ionization potential possess high coordination affinity with tannic acid to generate stable complexes [30]. According to HSAB, ionization potential is in the order of Fe^2+^ < Co^2+^ < Ni^2+^ < Cu^2+^ < Fe^3+^ < Zr^4+^ [53]. As a result, more metal content uptake in the complex were found in the cases of coordinating TA with Cu^2+^, Fe^3+^, and Zr^4+^ [30]. Regarding the complex stability constant of the TA and metal ion complex, TA-Cu^2+^ and TA-Ni^2+^ membranes were the most highly stable complexes with stability constants of 21.1 and 15.3, respectively [30].

Moreover, a study performed on the stability of TA-Fe^3+^ layers showed that films coated on a low-wettable surface are more stable than those coated on a highly wettable surface [54]. This finding has an important significance, considering the variation in the wettability of various membrane supports. However, the pH-responsive nature of polyphenol–metal assembly causes the stability of the films against solution acidity to be challenging. It can be seen from Figure 8b that for a membrane soaked in various solutions of distinct pH for 24 h prior to operation, flux increases significantly while rejection keeps declining along with increased acidity of the solution. This is attributed to the disassembly of the coordination under strong acidic conditions [14]. In contrast, the membrane performance is stable under basic conditions. This shows the assembled films’ resistance to alkaline feed solutions [7]. Fang et al. [32] further studied the release of iron ions from the assembled TA-Fe^3+^ complex membrane. In this study, it was confirmed that the amount of Fe^3+^ dissociating from the metal–polyphenol network increased with decreasing soaking solution pH. Again, no iron ion release was observed for membranes immersed in a solution of pH 5 or higher, verifying their durability under alkaline conditions.

## 6. Applications

The coordination-mediated self-assembly of polyphenol and metal ion is a novel strategy for fabrication of high-performance TFC membranes [24]. With their tunable average pore size [24,28], TA-Fe^3+^ membranes have great potential in various industrial applications involving liquid separation. These membranes have a promising potential in the ion or molecule separation field. They can also be used as a pretreatment in the desalination process.

Guo et al. [9] recommended non-polyamide TA-Fe^3+^-based membranes for wastewater reclamation and removal of organic contaminants. MPN-TFC NF membranes demonstrate a significantly high rejection property toward trace organic contaminants, especially endocrine disrupting compounds (EDCs) in wastewater. These membranes presented almost two orders of magnitude superior water/EDCs selectivity than commercial NF membranes of NF270, NF90, and XLE, though they had a relatively low salt rejection [9]. Hence, utilization of polyphenol-based membranes for wastewater reuse could be futuristic. TA-M^n+^ membranes are especially of high importance in the removal of pharmaceuticals, which are hardly removed from municipal wastewater by conventional treatment plants.

Metal–polyphenols network-based thin films are mainly exploited for fabrication of loose nanofiltration membranes with excellent dye/salt fractionation efficiency. Effective ion/organic selective separation performance was observed in a recent study alongside a more than 94% rejection toward various dyes of 320 to 1017 Da molecular weight [25]. Moreover, TA-Fe^3+^-based membranes possess a high flux property, as mentioned earlier. Ni^2+^–polyphenol-network-coated nanofiltration membranes also exhibited a favorable application for the treatment of dye in wastewater [44]. This presents a new approach to tackle the threat to global water safety from textile waste. Therefore, considering the dye desalination and concentration process to be one of the largest applications of NF membranes over the past two decades [55], the industrial demand for tannic-acid-based membranes could be promising. Moreover, MPN-based membranes show different properties toward dyes of different charge groups. High retention of anionic dyes can be achieved while maintaining low rejection for neutral dyes [56]. This is associated with the influence of Donnan exclusion caused by the negatively charged surface functional groups of the TA-Fe^3+^ selective layer. This allows for the selective fractionation and separation of dyes in waste streams. Nevertheless, the low salt rejection for NaCl should be given attention. Although with some optimization membranes of high salt rejection could be prepared [11], metal–polyphenol network membranes have commonly low salt rejection while more than 98% rejection could be achieved for many dyes and organic components. Chakrabatry et al. [45] especially portrayed the attractive purpose of these membranes to be various applications where organic solutes have to be desalted.

Another important feature of tannic acid is a noticeable antimicrobial and antifouling property. The coating of UF membrane with polyphenol and metal assembled film revealed to have decreased the interaction of membranes with microorganisms and proteins, mainly with negatively charged macromolecules including BSA and oils, providing excellent fouling resistance over the surface [27]. Moreover, tannic acid–metal ion coordination complexes enhance anti-bacterial and algal-inhibition properties of the ultrafiltration membrane [57]. TA-Cu-Fe layer coating over PVDF membrane introduces an excellent biofouling resistance ability, as shown in Figure 9e. The presence of tannic acid facilitates the reduction in Cu^2+^ in situ to Cu nanoparticles and enhances bacterial and algal growth inhibition efficiency [57]. This opens a wide range of applications in food, dairy, beverage, and protein purification industries.

Reverse osmosis (RO) is the most popular technology for the desalination of seawater and brackish water [58]. However, fouling, chlorine attack, and low water flux are still major challenges of the desalination process. TA-M^n+^ self-assembly has been extensively used for RO membrane modification in the immobilization of silver nanoparticles for enhanced antifouling property [58,59]. Poly(N-vinylpyrrolidone) (PVP) has also been immobilized on a metal–polyphenol precursor layer to improve the hydrophilicity and fouling resistance of RO membranes [60]. The TA-M^n+^ coating on the surface of RO membranes also facilitates removal of pharmaceutical contaminants from water, e.g., metronidazole [61].

Another important application of MPN-based membranes is oil/water emulsion separation. Oil/water emulsions are of environmental concern and require attention for resource recycling. Huang et al. [62] published a detailed review focusing on plant polyphenols (e.g., epigallocatechin, pyrogallol, and polydopamine) and their assemblies for membrane coating to separate oil–water emulsion. Song et al. [33] used a two-pot process to coat TA-Fe^3+^ on a polypropylene (PP) microfiltration membrane. The procedure involves adsorption of TA on the PP membrane followed by immersing the TA-loaded PP membrane in Fe^3+^ solution. This process resulted in a uniformly distributed coating of a TA-Fe^3+^ layer, which improved the wettability and water permeance of the membrane (Figure 9a,b). Tannic acid has been the most-used phenolic ligand for the fabrication of thin films of high surface wettability such as TA-Ti^4+^ [63,64], TA-Fe^3+^ [65], and TA-Ag^+^ [66] self-assembled layers. TA–metal ion coordination over the membrane surface converts the hydrophobicity of most polymeric membranes to hydrophilicity with underwater oleophobicity. The TFC membranes show not only reusability with a high flux-recovery ratio, but also desired stability for long-term operation [65]. An AgNPs-TA-PVDF microfiltration membrane maintained its chemical and mechanical properties after rinsing and scratch resistance tests (Figure 9d) [66]. Moreover, additives such as PEI [67] and MOFs [68] can be utilized in TA-M^n+^ self-assembly to improve the physicochemical properties of selective layers for emulsion treatment. In summary, this provides novel insights into the fabrication of high-efficiency membranes for oily water treatment. Furthermore, owing to their facile synthesis method, MPN-based membranes emphasize their potential practical application in the separation of oil/water emulsions.
Figure 9(**a**) Photographic images of a water drop (10 μL) on the membrane surface within 10 s after drop. (**b**) Water permeability and WCAs of superhydrophilic and underwater superoleophobic TA−Fe^3+^−coated polypropylene membrane [33] (copyright 2017, Elsevier). (**c**) Ag−NPs/tannic−acid−modified PVDF membrane for efficient separation of oil and water emulsion, (**d**) antifouling property of the Ag-NPs/tannic-acid-modified PVDF membranes during emulsion separation [66] (copyright 2022 Elsevier), and (**e**) anti-bacterial and anti-algal inhibition performance of TA−Cu^2+^−Fe^3+^−containing membranes [57] (copyright 2022, Elsevier).
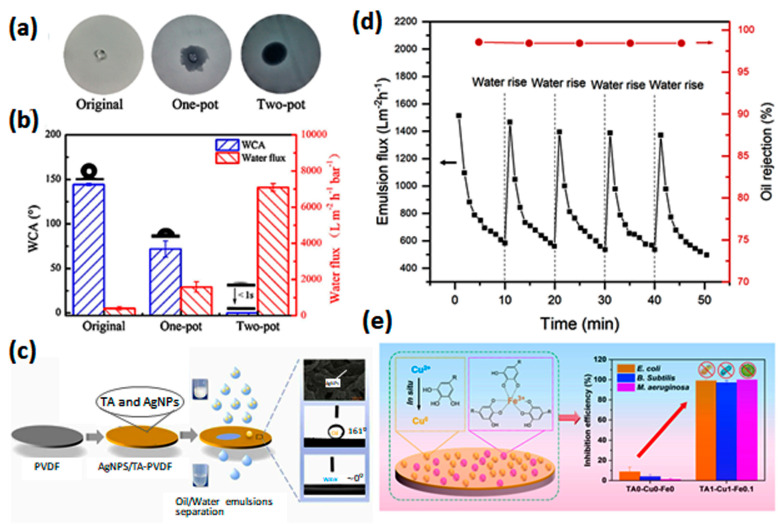


## 7. Future Prospects and Challenges of Polyphenol–Metal Assembly Membranes

Filtration techniques based on membrane technology have wide industrial and environmental application. The sustainable synthesis method, high water flux, and antifouling property makes MPN membranes attractive for practical applications in water and wastewater treatment. TA-M^n+^ thin films are ideal for a green strategy of fabrication of loose TFC NF and tight UF membranes. For efficient solute rejection, the following features should be possessed by NF membranes: a facile and fast synthesis method, high perm selectivity for energy and cost reduction, enhanced antifouling property, and stability under operating conditions. TA-M^n+^ membranes embrace these features. The pioneering findings in TA-Fe^3+^ coordination for film formation paved a way to synthesize superhydrophilic membranes of sufficient rejection in an utterly sustainable and green technique. However, the structural stability under a wide range of pH conditions is an important parameter for the industrial application of membranes. MPNs show disassembly in acidic conditions. Therefore, further studies focusing on the enhancement of the acid tolerance of the membranes is required to reaffirm their performance feasibility in liquid separation. The asymmetric porous supports such as PAN, PES, and PSF are commonly synthesized through phase separations: the non-solvent-induced phase separation (NIPS), thermally induced phase separation (TIPS), or vapor-induced phase separation (VIPS). NIPS is the most utilized technique for the fabrication of porous MF and UF membranes, and it basically relies on the practice of traditional organic solvents such as chloroform, *N*-methyl-2-pyrrolidinone (NMP) and dimethylformamide (DMF), and dimethylacetamide (DMAc) [3]. It is elucidated in this review that TA-M^n+^ TFC membranes are synthesized facilely, substituting the state-of-the-art polyamide TFC membranes. That being said, the fabrication of the porous support in an ecofriendly method is equally important. A rigorous assessment of the substitution of toxic organic solvents with green solvents for fabrication of porous membranes via green solvents is reviewed by Figoli et al. [3]. Moreover, solvent-free membrane fabrication has also been reported [69]. It is essential to integrate the ecofriendly synthesis of the porous support and MPN membranes for the total sustainability of the fabrication process.

Compared to existing polyamide membranes, metal–polyphenol-assembled membranes have superior rejection of hydrophobic trace organic compounds and contaminants [9]. However, much attention is needed in future studies to address the lower rejection of hydrophilic antibiotic compounds using loose NF membranes. Fine-tuning size-exclusion effects in these non-polyamide selective layers to realize high retention of both hydrophobic and hydrophilic trace organic contaminants is essential. Additionally, a membrane surface chemistry study is crucial for membrane–solute interaction regulated rejection enhancement.

Most assembly techniques prioritize polyphenol layer deposition over support substrates. This is associated with the adhesive property of the catechol-rich organic ligand. Metal ions are then added in solution form to crosslink with the already deposited polyphenol. However, some studies have been conducted in the reverse order by hydrolyzing membrane support first [7]. Hydrolysis creates negatively charged carboxyl groups at the substrate surface, which would regulate the distribution of positively charged metal ions over the surface via coordinative and electrostatic interactions. Nevertheless, deeper understanding of the complex formation and membrane structure, as well as comparative affinity of substrates to metal ions and tannic acid, is required. The effects of phenolic group or metal ion termination in the sequential deposition procedure on membrane microstructure and surface chemistry shall be investigated.

As stated earlier, the influence of the ionic strength of solutions on the complexation and deposition of metal–polyphenol networks was significant. However, the effect of reagent solutions salt concentration on polyphenol–metal self-assembly-based membrane morphology and performance is not studied yet. This information will endorse researchers to further improve the engineering of TA-M^n+^ membranes and foster membrane performance and efficiency. Moreover, though the stability tests performed showed a stable, as well as durable, coordination, the swelling properties of the membrane need investigation. It is noted that a decline in selective separation is exhibited in swelling membranes. Entry of small solvent molecules into the polymer network causes an expansion in distance between polymer chains, instigating a volume change. In this regard, PA membranes for nanofiltration and reverse osmosis are observed to swell in wet or liquid environments [70,71,72]. Further investigations are required to understand if metal–polyphenol self-assembly-based thin film membranes can overcome this challenge.

To realize effective separation of neutral or charged solutes via NF, a good understanding of the synergistic effects of partitioning mechanisms is required through modeling. A decent predictive model allows prediction of membrane performance and process optimization. Generally, modeling the complex membrane separations awaits to be addressed because of its lack of adequate research. An understanding of physical properties, as well as of physicochemical phenomena, inside NF pores is not well-established yet [52]. Therefore, future studies should focus on describing the transport of electrolytes and uncharged solutes through these NF membrane pores. An accurate but simplified model would allow us to understand the inherent mechanism for the transport or rejection of species and predict the membrane property during operation for multi-component salt or other solute solutions filtration.

Engineering of the metal–polyphenol complex films by implementing various methods of assembly to synthesize ultra-efficient membranes will lead to a major breakthrough in NF membranes. Further systematic research shall focus on understanding the transport of water and solutes in polyphenol-based materials. Development of membrane processes requires feasibility tests at several stages. Pilot-scale operations should be conducted to assess their technological viability. Long-term studies of TA-M^n+^ membranes on a lengthy period and large-scale process operation to determine their viability are required as well.

## 8. Conclusions

The problems of environmental pollution and the urge for the use of green solvents have sparked the development of sustainable membrane fabrication methods. Polyphenol-based nanofilms that are ecologically friendly and feasible could contribute in part to sustainable development and green chemistry. The metal-chelating mechanism of polyphenols allow the synthesis of thin-film composite membranes using solely aqueous solutions, eliminating solvent toxicity and hazardousness. TA–metal self-assembly-based membranes have only recently been introduced into the research industry, but their superhydrophilic nature has been shown to be of great importance. In this review, the performance and synthesis procedures of TA-M^n+^ membranes are discussed. Several parameters such as concentration, metal-to-ligand ratio, ionic strength, pH, and assembly time influences the thin layer synthesis, membrane pore structure, and membrane performance. TA–metal ion self-assembly-based membranes exhibit excellent water flux, rejection, and selectivity properties toward several solute components. Moreover, they possess outstanding membrane stability and antifouling property. MPN TFC membranes are extensively studied for water and wastewater treatment, including removal of pharmaceuticals, organic contaminants, dyes, dye/salt fractionation, desalination, and oil/water emulsion. Nevertheless, metal–polyphenol-based membranes have a relatively low rejection toward monovalent and divalent salts, as well as hydrophilic antibiotics, and show low structural stability in acidic conditions. Future studies shall focus on addressing these challenges. Furthermore, the rejection behavior of TA-M^n+^ TFC membranes toward different components should be thoroughly analyzed.

## Figures and Tables

**Figure 1 membranes-13-00481-f001:**
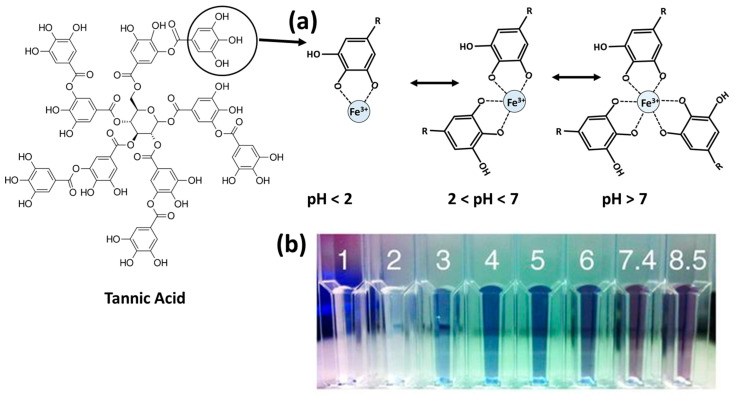
pH-dependent tannic acid–metal ion complexation; mono-, bis-, and tris-complex states (**a**). Mixtures have colorless, blue, and dark brown colors at the respective complex states. A photograph of the TA-Fe^3+^ capsule dispersions at the indicated pH values showing a color change of the TA-Fe^3+^ complexes in response to the pH of the solution (**b**) reprinted with permission from ref. [14]. © 2013, the American Association for the Advancement of Science.

**Figure 2 membranes-13-00481-f002:**
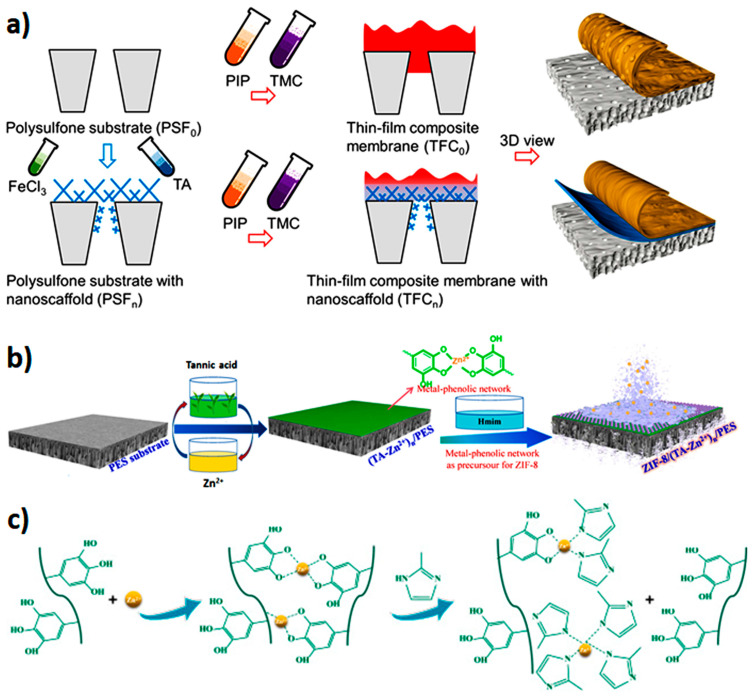
(**a**) Schematic diagram showing the synthesis of the conventional thin-film composite membrane (TFC0) and the TA-Fe^3+^ nanoscaffold-enhanced thin-film composite membrane (TFCn). To prepare the TFCn membrane, a polysulfone substrate was first coated with a TA/Fe nanoporous layer, followed by interfacial polymerization of PIP and TMC. The control membrane TFC0 was fabricated by skipping the TA-Fe coating procedure. Reprinted from ref. [42] with permission from American Chemical Society. (**b**) Schematic representation of the synthesis of ZIF-8/(TA-Zn^2+^)n/PES membrane with TA-Zn^2+^ network as interlayer and zinc precursor and (**c**) possible formation mechanism of ZIF-8 with a TA-Zn^2+^ network as metal resource reprinted from ref. [43] with permission from Elsevier.

**Figure 3 membranes-13-00481-f003:**
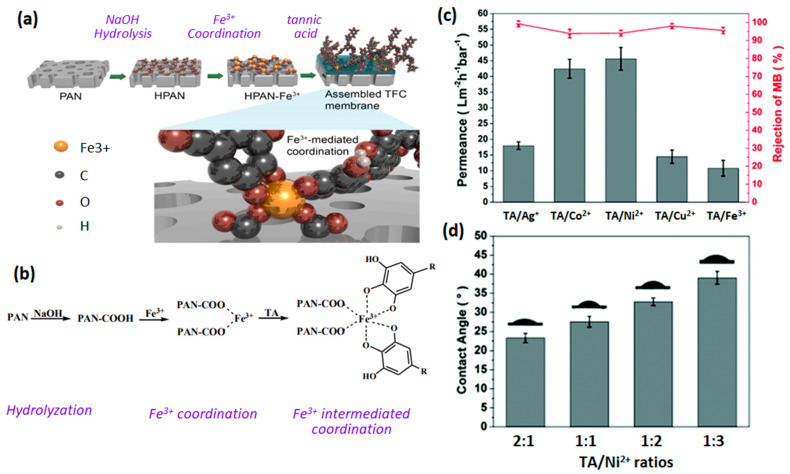
(**a**) Schematic diagram of the preparation process for the assembled TFC membranes and (**b**) reactions and the possible structure of the resultant Fe^3+^−intermediated TA thin film reprinted from ref. [7] with permission from Elsevier. (**c**) Separation performance of TA–metal ion−coated composite nanofiltration membranes, and (**d**) effect of TA/Ni^+^ ratio on water contact angle of membranes reproduced from ref. [44] with permission from the Royal Society of Chemistry.

**Figure 4 membranes-13-00481-f004:**
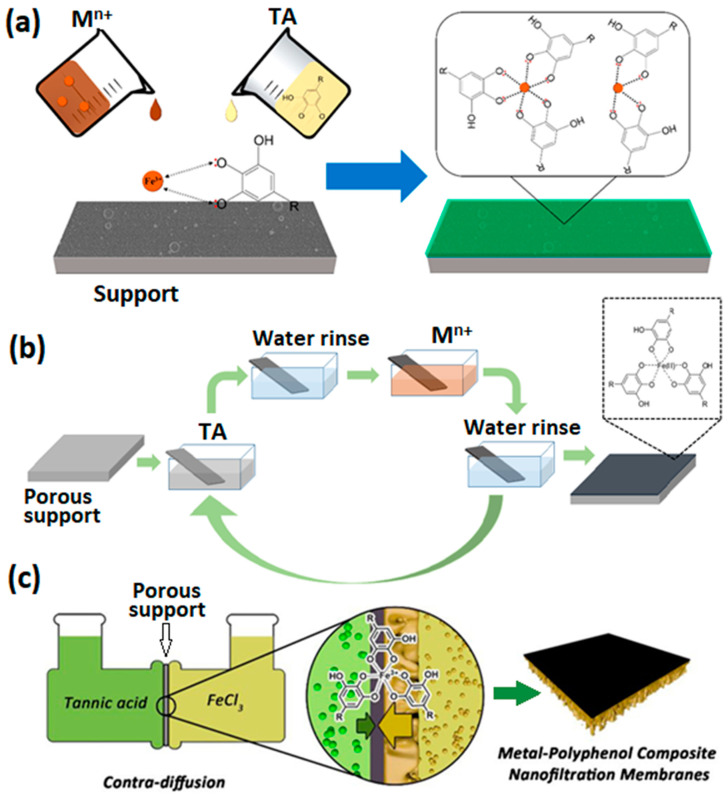
Schematic representation of metal–polyphenol NF membrane through (**a**) co-deposition. Reprinted from ref. [26] with permission from the American Chemical Society. (**b**) Layer-by-layer. In the LBL technique, synthesis of one layer of TA-M^n+^ membrane is analogous to a sequential two- or several-step dip-coating process. (**c**) the Fe^3+^/TA TFC membranes fabrication process via confined coordination by aqueous contra-diffusion. Reprinted from ref. [48] with permission from the American Chemical Society.

**Figure 5 membranes-13-00481-f005:**
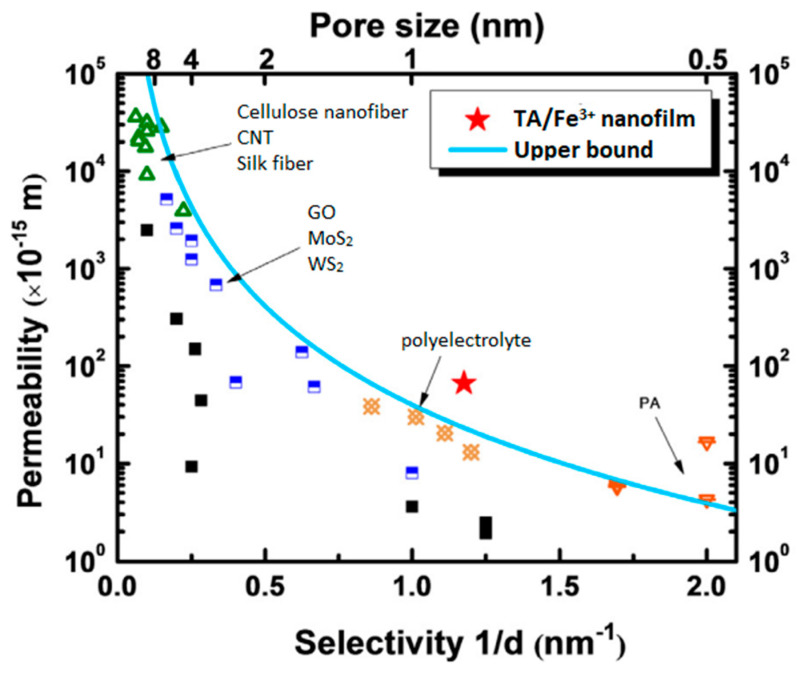
Comparison of the selectivity−permeability between the TA/Fe^3+^ nanofilm and other nanofilms made of various materials (carbon nanotube (green triangle), graphene oxide, molybdenum disulfide, and tungsten disulfide (blue and white square), polyelectrolyte (orange square), polyamide (orange triangle), other materials (black square)). Reprinted from ref. [28], ©2018, with permission from Elsevier.

**Figure 6 membranes-13-00481-f006:**
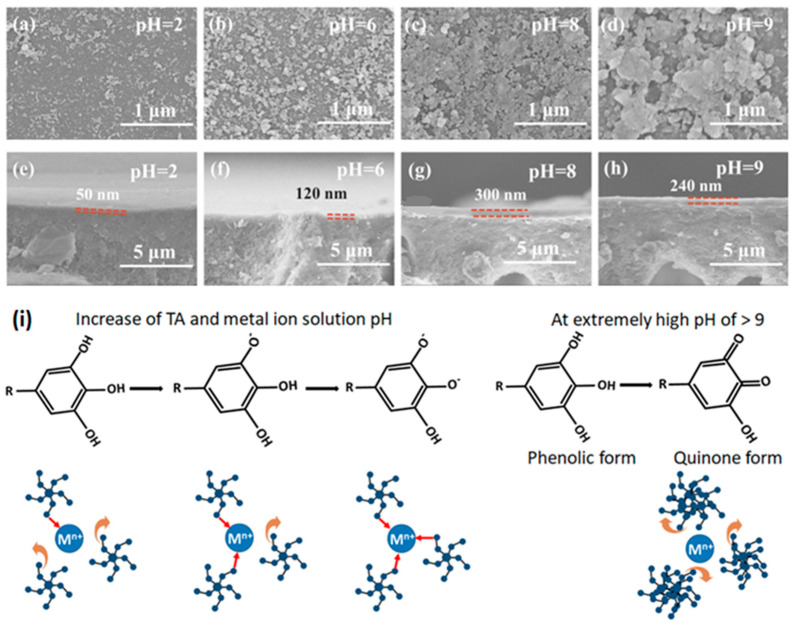
Surface and cross-sectional SEM images of membranes prepared under different buffer-solution pH values: (**a**,**e**) pH 2; (**b**,**f**) pH 6; (**c**,**g**) pH 8; and (**d**,**h**) pH 9. Reprinted from ref. [25] with permission from Elsevier. (**i**) Ionization of catechol groups of tannic acid in aqueous solutions of various pH levels and their respective coordination ability with transition metal ions. An increase in pH leads to the increase in crosslinking and the subsequent formation of thicker films with a decrease in surface porosity. However, at highly alkaline conditions, TA oxidation to quinone induces the formation of aggregate on the membrane surface.

**Figure 7 membranes-13-00481-f007:**
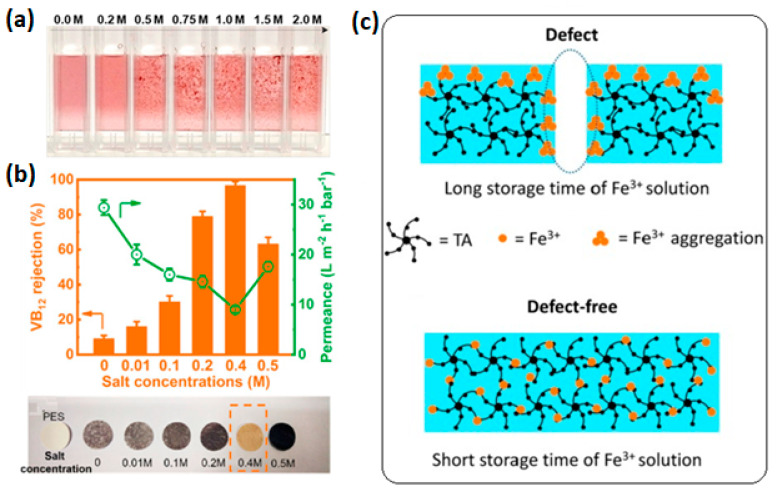
(**a**) Influence of salt concentration on the structure of the Fe^3+^-TA complexes in solution. Reprinted from ref. [51] with permission from the American Chemical Society. (**b**) Effect of salt concentration in aqueous solutions on the filtration performance and the appearance of PES and TA−Fe^3+^ composite membranes Reprinted from ref. [24] with permission from Elsevier, (**c**) Schematic illustration for the effect of Fe^3+^ aggregation on the TA/Fe^3+^ nanofilms. Reprinted from ref. [28] with permission from Elsevier.

**Figure 8 membranes-13-00481-f008:**
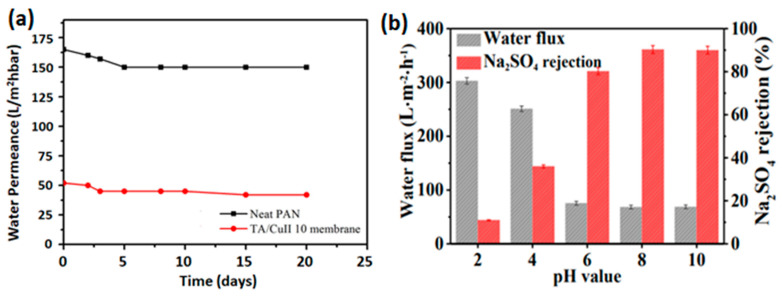
(**a**) Long-term continuous water permeance tests for neat PAN and TA−Cu^2+^ membranes [45] (copyright 2017 Elsevier) and (**b**) separation performance of (TA-Fe^3+^)_3_ membrane as a function of solution pH [7] (©2020 Elsevier).

**Table 1 membranes-13-00481-t001:** Separation performance of metal–polyphenol-coordination-based thin selective layers.

Membrane	Support	Preparation Method	Permeability (L∙m^−2^∙h^−1^∙bar^−1^)	Rejection (%)	Reference
TA-Cu^2+^	PAN	Co-deposition	52	Methyl orange: 65 Brilliant blue: 99	[45]
(TA-Fe^3+^)_5_ *	PAN	Layer-by-layer	40.9	93.9–100 toward various dyes	[25]
(TA-Zn^2+^)_5_	PAN	Layer-by-layer	32	Static blue carmine red: 60 Rose red sodium salt: 78	[25]
TA-Fe^3+^	Polyamide TFC	Co-deposition	7.52	Na_2_SO_4_: 99.5	[23]
TA-Fe^3+^	PES	Co-deposition	27.2	Congo red: 99	[8]
TA-Ni^2+^	P84	Co-deposition	45.6	Methyl Blue: 94.1	[44]
TA-Fe^3+^	P84	Co-deposition	9.8	Methyl Blue: 95	[44]
(TA-Fe^3+^)_2_	PES	Layer-by-layer	12.4	Methyl orange: 90 VB12: 98.9	[28]
TA-Fe^3+^	PES	Co-deposition	116	Heavy metals (Cu, Fe, Cd, Mn): 78–93	[27]
TA-Ti^4+^	PSF	One-step assembly	9	Methyl blue: 96.8 Congo red: 97.2	[47]
TA-Fe^3+^	Hydrolzed PAN	Co-deposition	13.6	Dyes: >99 Na_2_SO_4_: 90.2, MgSO_4_: 83.4	[7]

* subscripts represent the number of TA-M^n+^ self-assembled layers deposited over the support.

**Table 2 membranes-13-00481-t002:** Long-period filtration test of membrane stability.

Membrane	Support	Length of Operation	Performance	Reference
TA-Fe^3+^	PAN	72 h.	Only <7% permeability decline	[7]
TA-Fe^3+^	PES	30 days	Only 2% contact angle change	[32]
TA-Cu^2+^	PAN	20 days	Slight permeability change	[45]
TA-Fe^3+^	PES	24 h.	4.4% permeability decline	[8]
TA-Fe^3+^	PES	24 h.	No change in permeability and rejection	[24]
TA-Fe^3+^	PES/SPES	5 cycles	>82% flux recovery	[30]
TA-Fe^3+^	PAN	50 h.	3.3% flux decline whilst no change in rejection	[25]

## Data Availability

Not applicable.

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
