# Peer review of "Separation Performance of Membranes Containing Ultrathin Surface Coating of Metal-Polyphenol Network"

_membranes, 2023, doi:10.3390/membranes13050481_

Round 1
Reviewer 1 Report
1. In figure 1 title authors mentioned as "Figure 1. pH dependent tannic acid metal ion complexation; mono-, bis- and tris-complex states. Mixtures have colorless, blue and brown colors at the respective complex states." add suitable reference images to the description.
2. Figure 6 explains surface analysis at different pH levels, since authors mentioned this is one of the key study from this review its better to add furthers inputs like possible mechanisms and description. Which will get better reach and interest to the readers.
3. Figures quality should be improved.
Author Response
Reviewer #1:
Comments and Suggestions for Authors
- In figure 1 title authors mentioned as "Figure 1. pH dependent tannic acid metal ion complexation; mono-, bis- and tris-complex states. Mixtures have colorless, blue and brown colors at the respective complex states." add suitable reference images to the description.
Authors’ response:
We have added images of membranes fabricated at the above mentioned three complex states, and revised the manuscript accordingly.
- Figure 6 explains surface analysis at different pH levels, since authors mentioned this is one of the key study from this review its better to add furthers inputs like possible mechanisms and description. Which will get better reach and interest to the readers.
Authors’ response:
This section has been updated in response to the reviewers comment.
- Figures quality should be improved.
Authors’ response:
We have updated the figures according to the suggestion of the reviewer.
Reviewer 2 Report
This article provides a useful review of the research trend of membranes based on metal-polyphenol network, especially the fabrication methods and its characteristics.
It would be more informative to mention about the basic performence characteristics of the membranes referred to in the application section.
Similarly, membrane stability is very important for the application, but it would be nice to have the discussion not only on the solution conditions, but also on the applied loading condition and accelerated testing conditions. It would be very important to have the interpretation of the results from references on membrane stability, in view of application of newly developed membranes.
One point,
in the section 4., the author refers to "membrane pore structure". However, the SEM images in Fig.4 just show the cross-section of membranes, and there is no pore structure information. Please clarify how the author understand the pore structure of fabricated membranes reviewed, and give a concrete example for that if possible.
Author Response
Reviewer #2:
Comments and Suggestions for Authors
This article provides a useful review of the research trend of membranes based on metal-polyphenol network, especially the fabrication methods and its characteristics.
It would be more informative to mention about the basic performence characteristics of the membranes referred to in the application section.
Similarly, membrane stability is very important for the application, but it would be nice to have the discussion not only on the solution conditions, but also on the applied loading condition and accelerated testing conditions. It would be very important to have the interpretation of the results from references on membrane stability, in view of application of newly developed membranes.
Authors’ response:
If we understand this correctly, the reviewer is raising the issue of high pressure and mechanical stability discussions. Metal-polyphenol TFC membranes are thin layers deposited over support substrate. The issue of delamination at high pressure is already discussed in the paper that a strong adhesion confines the TA-Mn+ selective layer with the support. Under pressure extended period filtration investigations are also reviewed. However, to the best of our knowledge, other mechanical strength tests of these thin separation layers have not been reported. The mechanical strength of the thin film composite membranes come from the support layers.
One point,
in the section 4., the author refers to "membrane pore structure". However, the SEM images in Fig.4 just show the cross-section of membranes, and there is no pore structure information. Please clarify how the author understand the pore structure of fabricated membranes reviewed, and give a concrete example for that if possible
Authors’ response:
Section 4 discusses about the parameters influencing the metal-polyphenol self-assembly, and the resulting thin membrane selective layer characteristics and performances. The “Membrane pore structure” in the topic concerns about the increase or decrease in pore size, membrane surface porosity (dense or porous), layer thickness, and surface roughness relative to the studied parameters. The figure of membrane cross-section and top surface cited in this section (figure 6 and not “figure 4”) is an example of the effect of only one factor, i.e. pH. However, the concept of pore structure formation is beyond the scope of this review and is not or will not be discussed in this manuscript.
To avoid any ambiguity, we have renamed the section from ‘Factors affecting metal-polyphenol complex formation, membrane pore structure and performance’ to ‘Factors affecting metal-polyphenol complex formation, membrane selective layer characteristics and performance’.